An efficient parallel DCNN algorithm in big data environment

Mao Yimin 1
Nanehkaran Yaser Ahangari 2
Chandrasekaran Neelakandan 1
Huo Ying 1
Wen Zhan Qing 3
Gong ke 4
Decheng Miao deansgu@163.com 1
1 School of Information Engineering, Shaoguan University , Shaoguan City , Guangdong , China
2 School of Information Engineering, Yancheng Teachers University , Yancheng , Jiangsu , China
3 School of Information Engineering, Jiangxi University of Science and Technology , Ganzhou , Jiangxi , China
4 Jiangxi Branch, China United Network Communications Group Co., Ltd , Nanchang , Jiangxi , China
Angiulli Giovanni
Electronic publication date: 2025 May 20
Publication date: 2025
Volume: 11
Electronic Location ID: e2871
Received 2024 Oct 16; Accepted 2025 Apr 10
Copyright: ©2025 Mao et al.
Copyright year: 2025
Copyright holder: Mao et al.
License: This is an open access article distributed under the terms of the Creative Commons Attribution License, which permits unrestricted use, distribution, reproduction and adaptation in any medium and for any purpose provided that it is properly attributed. For attribution, the original author(s), title, publication source (PeerJ Computer Science) and either DOI or URL of the article must be cited.
License URL: https://creativecommons.org/licenses/by/4.0/

Keywords: Parallel DCNN, MapReduce, Im2col

Funding: The Key-Improvement-Projects of Guangdong Province 2022ZDJS048 SZ2022KJ06 Guangdong Natural Science Foundation 2021A1515011803 The Education Department Science and Technology Project in Jiangxi GJJ218505 GJJ218504 GJJ209406 This work was supported by the Key-Improvement-Projects of Guangdong Province (2022ZDJS048,SZ2022KJ06), Guangdong Natural Science Foundation under Grant (2021A1515011803), and the Education Department Science and Technology Project in Jiangxi (GJJ218505, GJJ218504, GJJ209406). The funders had no role in study design, data collection and analysis, decision to publish, or preparation of the manuscript.

==============================
Big data plays a vital role in developing remote sensing, landslide prediction, and enabling applications, the integration of deep convolutional neural networks (DCNN) has significantly improved its prediction accuracy. However, several challenges remain in processing vast satellite imagery and other geospatial data. These challenges include excessive redundant features, slow convolution operation, and poor loss function convergence. An efficient parallel DCNN algorithm (PDCNN-MI), combined with MapReduce and Im2col algorithms, is introduced to address these challenges. First, a parallel feature extraction strategy based on the Marr-Hildreth operator (PFE-MHO) is proposed to extract target features from data as inputs to the network, effectively solving the problem of high data redundancy. Next, a parallel model training strategy based on Im2col method (PMT-IM) is designed to remove the redundant convolutional kernels by designing the center value of distance, improving convolution operation speed. Finally, a small batch gradient descent strategy (IMBGD) is presented to exclude the influence of training data of anomalous nodes on the batch gradient and solve the problem of poor convergence of the loss function. By utilizing these enhancements, the experimental results indicate that PDCNN-MI outperforms existing algorithms in classification accuracy and is well-suited for fast and large-scale image dataset processing.

Introduction

Big data has become a pivotal subject in the modern information era, influencing economic trends, societal advancements, and the development of cutting-edge technological research and innovation. Among the many fields transformed by big data, remote sensing has experienced a significant revolution due to the exponential growth of geospatial data. Advances in data acquisition technologies, such as high-resolution satellites, Light Detection and Ranging (LiDAR) (Tolan et al., 2024), and uncrewed aerial vehicles (UAVs) have enabled the collection of detailed terrain and environmental data essential for predicting landslides. Landslides pose severe threats to human life, infrastructure, and economic stability, particularly in mountainous and geologically unstable regions. Therefore, the accurate and timely prediction of landslides is vital for effective disaster risk management, enabling the implementation of proactive measures to protect lives, property, and ecosystems.

Advanced analytical methodologies are essential to utilize large-scale remote sensing datasets effectively. However, the complexity and scale of remote sensing data, characterized by enormous volume, high dimensionality, and multi-source heterogeneity, present significant challenges for traditional analytical techniques. Conventional methods, which typically depend on manual feature engineering and simplified models, struggle to capture intricate spatial and temporal patterns of such data. As a result, these methods often fail to achieve the accuracy and robustness required for reliable landslide prediction, especially when processing large-scale, high-resolution datasets that include diverse environmental and geological variables.

To address these challenges, deep convolutional neural networks (DCNN) have emerged as powerful tools. DCNN can automatically learn hierarchical features from raw data, eliminating the need for manual feature extraction and enabling more precise modelling of complex geospatial relationships. This capability has resulted in state-of-the-art performance in feature extraction, terrain analysis, and landslide susceptibility mapping.

Although DCNNs have achieved remarkable success in remote sensing applications, their deployment in big data environments is constrained by inherent limitations that are exacerbated by geospatial data’s massive volume, high dimensionality, and heterogeneous sources. Three critical challenges are identified: (1) Excessive redundant features: High inter-band correlations and background noise in multi-source remote sensing data create significant feature redundancy. Traditional DCNNs lack dynamic feature selection mechanisms, causing redundant information to accumulate across layers. This accumulation increases computational load, heightens the overfitting risk, and ultimately degrades model generalizability. (2) Slow convolution operations: Conventional convolution operations involve repetitive computations and inefficient memory management when processing large-scale images. Their high computational complexity hinders effective parallel processing in distributed environments, resulting in underutilized hardware resources and limited training efficiency. (3) Poor loss function convergence: In high-dimensional feature spaces, interference between noisy and effective gradients—coupled with the adverse impact of outlier samples on parameter updates—prevents traditional optimization algorithms from converging reliably. Consequently, models are prone to getting trapped in local optima and experiencing significant training oscillations.

In response to the challenges outlined above, this paper introduces a parallel DCNN algorithm (PDCNN-MI), which combines the MapReduce distributed computing framework with Im2col techniques. By harnessing the parallel processing power of MapReduce, the algorithm significantly improves the training and inference efficiency of DCNNs. Furthermore, the framework’s scalability effectively addresses the challenges associated with large-scale datasets. This method paves the way for more efficient, robust and scalable landslide prediction systems in big data environments. The core contributions of the PDCNN-MI algorithm include the following three aspects:

(1) Parallel feature extraction strategy based on Marr-Hildreth operator (PFE-MHO)

An efficient parallel feature extraction strategy, PFE-MHO, is proposed, which significantly enhances the efficiency and quality of feature extraction through a multi-level feature screening mechanism. Addressing the issue of excessive redundant features in traditional methods, PFE-MHO dynamically filters and optimizes features, effectively reducing the number of redundant features. This not only lowers data storage and computational costs but also improves the representativeness of features and the quality of model inputs.

(2) Parallel model training strategy based on Im2col method (PMT-IM)

A parallel model training strategy, PMT-IM, based on the Im2col method, is introduced. By optimizing convolution kernels and leveraging distributed computing, it significantly accelerates convolution operations and improves the efficiency of model training. To tackle the problem of slow convolution operations, PMT-IM transforms convolution operations into matrix multiplication using the Im2col method. Combined with the parallel computing capabilities of MapReduce, it greatly reduces the time complexity of convolution operations. Additionally, by optimizing convolution kernels using the Mahalanobis distance central value (MDCV), it further minimizes computational redundancy.

(3) Improved mini-batch gradient descent strategy (IMBGD)

An improved mini-batch gradient descent strategy, IMBGD, is proposed, which enhances the stability and convergence speed of model training through outlier node detection and adaptive learning rate adjustment mechanisms. Addressing the issue of poor loss function convergence, IMBGD dynamically identifies and mitigates the impact of outlier nodes. Combined with a multi-level gradient correction mechanism, it effectively resolves oscillations and slow convergence during the gradient descent process, ensuring efficient training and stable performance of the model on large-scale datasets.

The remainder of the paper will comprehensively explore the proposed PDCNN-MI algorithm and its contributions. ‘Related Work’ reviews current approaches to parallelizing DCNN training and critically examines their limitations. ‘Preliminary’ introduces key concepts, including non-local means, cosine similarity measures, Im2col, and Mahalanobis distance, which form the foundation of the proposed methodology. ‘The Proposed Method’ discusses the technical aspects of the PDCNN-MI algorithm, focusing on its innovative strategies for feature extraction, model training, and gradient descent optimization. ‘Experimental Evaluation’ presents a thorough analysis of the experimental results, comparing the performance of the algorithm with state-of-the-art methods and evaluating its effectiveness in big data environments. ‘Applications’ briefly discusses the application of PDCNN-MI in landslide prediction. Finally, ‘Conclusions’ concludes the paper, summarizing the key findings and suggesting potential avenues for future research.

Related Work

Deep convolutional neural networks (DCNN) (Adegun, Viriri & Tapamo, 2023) are powerful classification algorithms in deep learning. They are known for their shift-invariant properties, strong generalization capabilities, and efficient feature extraction. These characteristics make DCNN particularly effective in enhancing image feature extraction. For instance, Yan et al. (2023) introduced an improved multilayer perceptron (CNN-MLP) that combines the strengths of both CNN and augmented multilayer perceptron (MLP) techniques. This model utilizes data augmentation to enlarge the training datasets and employs an Adagrad-based MLP optimizer as a depth classifier for classifying remote sensing images. Experimental results demonstrate that the model outperforms existing methods in classification accuracy.

Similarly, Wang & Wang (2023) proposed the CNN-CapsNet model for remote sensing image scene classification. This model uses a DCNN pre-trained on ImageNet as a feature extractor and then maps the extracted features to a modified CapsNet for classification. Experimental results show that this approach delivers strong performance. However, with the rapid development of mobile Internet, there has been exponential growth in multimodal and high-value data in remote sensing technology. This growth has led to challenges, such as long model training times and repeated parameter updates due to the diversity of data. As a result, addressing the training costs of DCNN models in big data environments has become increasingly important.

Google’s MapReduce framework provides significant benefits in computing resource scheduling and rapid processing of big data (Rozony et al., 2024; Kang, Pan & Liu, 2022; Guo, Huang & Hou, 2022). It operates on a distributed data store and facilitates parallel processing of large clusters. Distributing the DCNN model across MapReduce nodes for parallel training can significantly improve computational efficiency compared to traditional single-machine methods.

Several approaches have been developed to address the challenges associated with big data analysis and processing using deep learning techniques. For instance, Asadianfam, Shamsi & Rasouli Kenari (2021) proposed a deep neural network architecture that integrates MapReduce, a core component of Hadoop, for image labeling and classification. This study represents the first use of an MR-LSCNN structure for processing traffic images. Utilizing a dataset of 10,000 images of offending drivers from a traffic control center, the proposed structure demonstrated impressive classification accuracy. The approach segments input images to extract essential features such as objects, locations, resources, surrounding objects, and actions, thereby highlighting the potential of deep learning architectures in domain-specific applications. The segmentation process, while effective, could compromise computational efficiency due to excessive redundant features.

Subsequently, Geetha et al. (2024) proposed a hybrid deep learning model that integrates DCNN with an enhanced DeepMaxout classifier, known as the N-Sigmoid Activated Maxout-Convolutional Network (N-SAMCN), for crop disease classification. This approach enhances image classification accuracy compared to traditional classifiers. However, despite its improved feature extraction and classification performance, N-SAMCN faces challenges such as high computational cost, increased risk of overfitting, and slower inference speeds, making the training process more complex. These limitations are exacerbated, particularly when handling high-resolution agricultural imagery, by slow convolution operations.

To further enhance the efficiency and scalability of big data classification, Chidambaram, Cyril & Ganesh (2023) proposed the Elastic Collision Seeker Optimization-based Faster R-CNN (ECSO-FRCNN) classifier. This method combines the Elastic Collision Seeker Optimization (ECSO) algorithm with Faster R-CNN to deliver superior classification performance. The ECSO-FRCNN classifier effectively addresses critical challenges such as missing attributes and incremental learning by transforming input datasets into a probability index table and determining the membership degrees for each data attribute. By leveraging the MapReduce framework, this approach significantly enhances training performance and scalability, making it well-suited for large-scale data processing tasks. However, the reliance on probabilistic transformations may lead to suboptimal outcomes, especially in scenarios with imbalanced or noisy datasets, due to poor loss function convergence.

Preliminary

Non-local means

Non-local means (Zhou et al., 2023) is a filter based on adjacent pixels, denoising data using the image’s global information. For a given data sample U, the grey scale value u¯x Filtered by nonlocal means, it can be expressed as below. (1) u¯x= ∑y∈Gϕx,y∗θy

where x denotes the search window, y denotes the adjacent window, ϕ(x, y) is the region similarity between x and y, and θ(y) is the noisy image.

Cosine similarity

Cosine similarity (Eminagaoglu, 2022) is a standard measure of similarity between two images. It maps the data of both images into vector space and then calculates the cosine of the inner product space between the vectors to compare the similarity. Assuming that X, Y denote the comparison of individuals, and x, y denote the one-dimensional form of the individuals, the cosine similarity Sim(X, Y) is expressed as follows. (2) SimX,Y=cosθ=x¯.y¯x+y

where x,y is the mode of the image vectors.

Image to column

Image to column (Im2col) (Li et al., 2023) is a transformation function that converts convolutional operations into matrix multiplication. It reshapes a 3D input data matrix into a 2D matrix and transforms the convolution kernel into a 1D matrix, simplifying the convolution process. Assuming that x is the input feature map, w is the convolution kernel, and i, j are the vertices, the convolution result can be expressed as follows. (3) ai,j= ∑h=0kH−1 ∑w=0kW−1wh,wxi+h,j+w

where kH, kW is the sum of the size of the image.

Mahalanobis distance

Mahalanobis distance (Li et al., 2024) is a commonly used distance metric that effectively measures the similarity between two unknown sample sets. Suppose that x represents the covariance matrix of a multidimensional random variable, the sample mean, and the current single-point data. Then, the martingale distance is as below. (4) DMx=x−μTS−1x−μ.

The Proposed Method

The PDCNN-MI algorithm is proposed to address the challenges of excessive redundant features, slow convolution operation, and poor loss function convergence in remote sensing image classification. It consists of four stages: data enhancement, feature parallel extraction, model parallel training, and parameter parallel updating.

In the data enhancement stage, rotation, flipping, and Gaussian noise are utilized to address imbalanced data distribution. The class imbalance is mitigated while dataset diversity is enhanced, with key features preserved. Minority-class representation is effectively improved, leading to more balanced model training.

In the feature parallel extraction stage, the PFE-MHO strategy begins by removing noise using a non-local mean filter based on cosine similarity. Next, features are extracted through a Laplacian operation, and finally, a feature correlation index is applied to screen out redundant features. This process improves data quality, reduces the impact of irrelevant features on training accuracy, and minimizes computing resource overhead.

In the model parallel training stage, the PMT-IM strategy calculates the convolution kernel covariance matrix and its mean value to construct the Mahalanobis distance center. It then clips redundant convolution kernels and employs the Im2col algorithm to convert convolution operations into matrix operations, enabling parallel processing that accelerates convolution layer operations.

During the parameter parallel updating phase, the IMBGD strategy calculates the mean weight of the loss and the summation of loss gradients to construct a batch gradient. This approach mitigates the impact of anomalous node data and improves the convergence of loss function. Collectively, these strategies optimize the overall performance of the algorithm. The structural workflow of the PDCNN-MI algorithm is illustrated in Fig. 1.

Figure 1 The flow chart of PDCNN-MI algorithm.

Data enhancement

To address the inherent class imbalance in the dataset, data augmentation techniques are employed to enhance the generalization capability of trained models. The methodology consists of three principal transformations:

Image rotation: Given an image sample I(x, y) with spatial coordinates (x, y), the rotated image I′ is computed as: (5) I′x,y=IRx,y

(6) R=cosθ−sinθ sinθ cosθ

where R is rotation matrix, θ is uniformly sampled from ± 15° to preserve structural integrity while introducing meaningful variations.

Bidirectional flipping: To maximize data utility while maintaining semantic consistency, comprehensive flipping operations are implemented in both spatial dimensions. The transformations are precisely defined as: (7) Ihflipx,y=Iw−xyIvflipx,y=Ixh−y.

Among them, Ihflip refers to horizontal flipping, Ivflip refers to vertical flipping, w is the image width, and h is the image height.

Gaussian noise: To improve model robustness against sensor noise and acquisition variations, controlled noise augmentation is performed: (8) Inoisyx,y=Ix,y+N0,σ2.

Here, σ is the standard deviation of Gaussian noise, and σ is set to 0.05.

This systematic augmentation approach significantly increases minority class representation while maintaining the statistical properties of the original dataset. Each transformation is carefully designed to preserve label correctness and feature integrity, ensuring the augmented data remains physically plausible and semantically meaningful for training. The combined transformations effectively address class imbalance while enhancing model generalization capabilities.

Feature transformation function

A novel PFE-MHO strategy is proposed to eliminate redundant features in the image data. The strategy consists of two steps: feature extraction and feature selection.

Feature extraction: To obtain high accuracy of image features, we apply a non-local mean filter FT(a, b) based on cosine similarity, to be remove noise from the initial dataset. The process is as follows: (1) the adjacent window matrix and search window matrix centered on the pixel are set up in the target image, and the adjacent window is sliced in the current image. (2) the weighted values of the adjacent window are obtained by comparing the cosine similarity of the matrix where the pixel points a, b are located. (3) the convolution kernel f(x, y) of size is set to 3*3, and Laplace operations on g(x, y) is performed to obtain the equation h(x, y) = ∇2(f(x, y)⋅FT(a, b)). (4) If the second-order derivative of the Laplace equation for the node is cross-zero and the first-order derivative is at a large peak, the node is retained. Otherwise, the pixel is set to zero; (5) the current data nodes are merged to get the image after feature extraction.

Theorem 1. Nonlocal mean filter FT(a, b). Assuming a, b represent the adjacent window centered on a pixel a andthe search window centered on the pixel b respectively. The transformation function FT(a, b) is calculated as follows. (9) FTa,b= ∑b∈Gia¯.b¯a+b⋅θa

where θ is the noisy image, and Gi is the current image data.

Proof. If the value of the noise-free pixel block is ω(x, y) and the noise value is ψ(x, y), then the value of the pixel block fused with the noise is ρ(x, y) = ω(x, y) + ψ(x, y), the mean value of the pixel block is ρ ¯x,y=1k⋅∑i=1kρix,y, and the expected value is Eρ ¯x,y=1k⋅∑i=1kEωix,y+Eψx,y. Due to the similarity between pixel blocks, E[ωi(x, y)] can be simplified to ω(x, y). When the noise is 0, E[ψ(x, y)] = 0, so it follows that Eρ ¯x,y=ωx,y. In addition, due to the non-correlation of the noise, the variance of ω(x, y) is σρ ¯x,y2=σωx,y2+1k2∑i=1kσψi22. Since ω(x, y) is noiseless and corresponds to a variance of 0, there is σρ ¯x,y2=1k⋅σψx,y2. This suggests that the noise ψ(x, y) is correlated with the variance and FT(a, b) reduces the data noise by reducing ψ(x, y).

Feature selection: After the feature extraction is completed, the images in the batch are divided into smaller chunks, and redundant features are eliminated by calculating the feature similarity between image blocks. The specific process is as follows: (1) the same class of images is sliced into equal size blocks and the feature correlation index FCI(x, y) to calculate the similarity between two image blocks x, y. (2) the key-value pairs <(x, y), FCI(x, y) > are stored in HDFS and remove redundant features in the image by removing items with FCI(x, y) < ɛ in the key-value pairs according to the correlation coefficient ɛ. (3) the key-value pairs are traversed again, and the key of the remaining key-value pairs in HDFS is read to obtain the number of the image block after filtering.

Theorem 2. Feature correlation index FCI(x, y). Suppose that x, y represent the two feature vectors, μx, μy represent the expectation of x and y, and σx, σy represent the variance of x and y. The formula for calculating the correlation index FCI(x, y) is as follows. (10) FCIx,y=2σx⋅σyσx2+σy2+μx−μy2.

Proof. FCI(x, y) is a measure of the feature similarity between x and y. Set μx, μy denote the expectation of x and y, and σx, σy denote the variance of x and y. When σx = 0, the convolution operation is a linear superposition, and FCI(x, y) = 0; When σx ≠ 0, σy ≠ 0 and the features of the feature vectors x and y are similar, FCI(x, y) → 1.

Model parallel training

The PMT-IM strategy, which consists of two main stages, convolution kernel pruning and parallel Im2col convolution, is proposed to eliminate redundant convolutional kernels during the execution of parallel convolutional operations.

Convolution kernel pruning: To reduce the invalid computation caused by redundant convolutional kernels, the MDCV system is designed to sieve out invalid convolutional kernels. The specific process involves: (1) the covariance matrix S and mean µ of all the convolution kernels calculated in the convolution layer to construct the objective function f(x) of the MDCV. (2) the second-order Taylor expansion f(x) = xk + ∇f(xk)(x − xk) + 0.5⋅(x − xk)T∇2f(xk)(x − xk) of f(x) is computed at its stationary point xk. (3) if the current second-order derivative is non-singular, the next iteration point is xk+1 = xk + ∇2f(xk)−1∇f(xk), otherwise the ∇2f(xk)d =  − ∇f(xk) is calculated to determine the search direction dk, and then the next iteration point xk+1 = xk + dk until the optimal MDCV value is found. (4) The distance dist between all the convolutional kernels and the MDCV value is calculated, and the convolutional kernels dist < α are pruned to complete the convolutional kernel pruning process.

Theorem 3. Mahalanobis distance center value MDCV. X1, X2, …, Xn represents the convolutional kernels in the network model, S represents the covariance matrix of the convolutional kernels, and µ represents the mean of the convolutional kernels. The equation for the MDCV of the Mahalanobis distance centroid is as follows. (11) MDCV=x∗=min∑x∈Rnx−μTS−1x−μ.

Proof. Suppose S is the covariance matrix of the vector group X1, X2, …, Xn and µ is the mean of the vector group. S is used to exclude the interference of similarity between variables. MDCV is the minimum distance from the eigenvector x∗ to the eigenvector group X1, X2, …, Xn. When the feature vector x → MDCV, the feature vector x is replaced easily by X1, X2, …, Xn. When it means that x is linearly correlated with X1, X2, …, Xn. Therefore, MDCV is the minimum distance that indicates x∗ to X1, X2, …, Xn.

Parallel Im2col convolution: The parallel Im2col convolution is combined with the MapReduce computing framework to extract image features. The main components are as follows. (1) Im2col method is used to map the feature map Mi into a convolutional operation matrix Ii, and the key-value pair <Ii, Kz > is stored with the corresponding convolutional kernel; (2) Map() function is used to perform data allocation; (3) perform matrix multiplication operation between Ii and the one-dimensional vector of the corresponding convolutional kernel to obtain the intermediate convolution result; (4) Reduce() function is used to merge the feature maps of the same data to obtain the output feature map.

Parameter parallel updating

To solve the problem of poor loss function convergence caused by abnormal nodes in the training data during gradient descent, an IMBGD strategy is proposed. This consists of gradient construction and parameter parallel updating.

Gradient construction: To eliminate the effect of abnormal node training data on batch gradient, loss of mean weight LAW(gi) and loss summation gradient LSG(T) are designed. The concrete process is: (1) the mean value of the loss function of batch data is calculated according to the formula of the loss function, and the difference between the loss function and the mean value is calculated to obtain the loss mean weight, which is mapped to the key-value pair <gi, LAW(gi) > and stored in HDFS. (2) the partial derivative∇Jδiof the loss function of each data gi is calculated concerning the current parameter δz, and the result <gi, ∇Jδi > is stored in the HDFS. (3) the <gi, LAW(gi) > and <gi, ∇Jδi > is traversed with gi as the index, the average gradient LSG(T) is constructed, and the batch gradient of the current parameter is obtained.

Theorem 4. Loss mean weight LAW(gi). Suppose gi denotes a piece of data in a batch, J(ω, b)i denotes the loss function value of the data gi, batch_size is the batch data size, and LAD(gi) is the absolute value of the difference between J(ω, b)i and the average of the loss function value. The formula for calculating the loss-mean weight LAW(gi) is as follows. (12) LAWgi=1LADgi<τ0LADgi≥τ

(13) LADgi=∑i=1batch_sizeJω,bi/batch_size−Jω,bi.

Proof. LAW(gi) is the weight indicator of the loss function value of data gi.τ is the threshold value to measure LAD(gi). When LAD(gi) < τ, gi belongs to the regular value, LAW(gi) = 1. When LAD(gi) ≥ τ, gi belongs to the abnormal value, LAW(gi) = 0. Therefore, different values of LAW(gi) can be used to decide whether to keep the loss function value of gi or not.

Theorem 5. Loss summation gradient LSG(T). Assume that T denotes all data in the batch, ∇Jxi denotes the gradient of the loss function of data gi with respect to parameter x, and batch_size denotes the batch data size. LSG(T) is calculated as follows. (14) LSGT=∑i=1batch_size∇Jxi×LAWgibatch_size

Proof. It is known that ∇Jxi is the gradient of the loss function of gi for parameter x, and batch_size is the batch data size. When LIW(gi) = 1, the gradient ∇Jxi of data gi decreases toward the optimal direction. When LIW(gi) = 0, the gradient ∇Jxi of data gi deviates more from the optimal direction and is not counted in LSG(T).

Parameters updating: The parameters are updated using the propagation algorithm in the following error term procedure to obtain the model after parallel update of parameters. The process is as follows: (1) the gradients ∑i=1kLSGTl−1 of all parameters of the convolution kernel Wkl−1 at layer l − 1 are computed and the results are mapped as key-value pairs <Wkl−1,∑i=1kLSGTl−1> into HDFS. (2) the change ΔWkl−1 of the parameters of the convolution kernel Wkl−1 is calculated, and the network parameters of the convolutional kernel at the layer l − 1 are updated. (3) the updated parameters are synchronized to all nodes via HDFS, and then the next update is performed until all parameters in the model are updated.

Time complexity

To effectively analyze the time complexity of the PDCNN-MI, the algorithms MR-LSCNN, ECSO-FRCNN , and N-SAMCN are selected for comparison.

The time complexity of the PDCNN-MI algorithm is primarily composed of three parts: feature extraction in parallel, model training in parallel, and parameter update. The time complexity of each section is calculated as follows.

(1) Feature parallel extraction

The time complexity of this stage consists of two main components: feature extraction and feature screening. Suppose the number of samples is n, the number of cluster nodes is k, and the maximum number of Laplace iterations is m. In the feature extraction stage, two sliding windows are used to traverse the data and the cross-zero of the target window a is calculated, thus the time complexity is Om⋅n⋅lognk. In the feature screening stage, the time complexity of calculating the similarity of two data slices is O(n2). Thus, the total time complexity is as follows. (15) T1=Om⋅n⋅lognk+n2.

(2) Model parallel training

The time complexity of this stage includes two components: the computation of the MDCV value and the parallel convolution operation. Assuming the number of convolution kernels is p, and the size of the convolution kernel is s, the algorithm calculates the standard deviation of the convolution kernels and identifies the maximum linearly correlated kernel in the process of computing the MDCV value. This has a time complexity of Op⋅s2k. After the data processing stage in the previous phase, the algorithm only needs to perform matrix multiplication during the parallel convolution operation, which has a time complexity of O(n2). Therefore, the time complexity of model parallel training is given as follows. (16) T2=Op⋅s2k+n2.

(3) Parameter parallel updating

A gradient construction method for batch data is proposed in this stage. Assuming the number of nodes is k and the size of the fully connected input to the model is c, the time complexity of batch data gradient construction is O(k⋅c2). Therefore, the time complexity of the parameter parallel update is as follows. (17) T3=Ok⋅c2.

In a word, the time complexity of PDCNN-MI algorithm is as follows. (18) TPDCNN−MI=T1+T2+T3=Om⋅n⋅logn+p⋅s2k+n2+k⋅c2.

In the MR-LSCNN algorithm, spatial–temporal features are processed sequentially through convolutional operations and LSTM-based temporal modeling. Assuming p is the number of convolutional kernels, q is the kernel size, and h is the LSTM hidden units, the time complexity is as follows: (19) TMR−LSCNN=On⋅p⋅q+n⋅h2.

In the ECSO-FRCNN algorithm, the Faster R-CNN backbone and ECSO-based optimization are jointly applied. Assuming c is the number of convolutional kernels in the CNN backbone, q is the average kernel size, p is the number of region proposals, and k is the iteration count of ECSO, the time complexity is formulated as: (20) TECSO−FRCNN=On⋅C⋅q+p⋅q+k⋅p+logp.

In the N-SAMCN algorithm, the adaptive bilateral filtering, multi-feature extraction and deep hybrid classifier are jointly applied. Assuming n is the pixel count, k is the filter size, r is the LGXP neighborhood radius, d is the GLCM gray-level, c is the feature channels, and h is the hidden units, the time complexity is formulated as: (21) TN−SAMCN=On⋅k2+n⋅r2+d2+c2⋅k2+c3+n⋅h.

Since the value of n is much larger than other metrics in the big data environment, the time complexity of the PDCNN-MI algorithm is smaller than other algorithms, namely, TPDCNN−MI < min(TMR−LSCNN, TESCO−FRCN, TN−SAMCN).

Experimental Evaluation

This section describes the hardware and software environment used for the experiments and evaluates the optimization effects of the proposed PDCNN-MI algorithm on model training. The assessment is conducted through comparative experiments, which provide insights into the performance improvement of the algorithm.

Experimental settings

The experimental setup consists of a computing cluster comprising eight nodes, including one master node and seven slave nodes, with their configurations detailed in Table 1. Each node has an Intel i5-12400F processor, 16GB RAM, and an NVIDIA RTX 3070Ti graphics card with 16GB of memory. To ensure efficient data transmission, the nodes are interconnected via a 1Gb/s Ethernet network. The software environment for the experiments includes Ubuntu 22.01 as the operating system, Python 3.9 for scripting and model implementation, JDK 18.0.1 for Java-based processes, and Hadoop 3.2.1 for distributed computing.

Table 1 Node configuration information.

Node type	Node name	IP	
Master	Master	192.168.105.1	
Slaver	S1∼S7	192.168.105.2∼8	

Experimental dataset

To evaluate the effectiveness of the PDCNN-MI algorithm, four datasets from different domains were used in the experiments: EMNIST-Balanced (Baldominos, Saez & Isasi, 2019), CompCars (Xu et al., 2022), ImageNet 1K (Li et al., 2022), and COCO5 (Lin et al., 2014). Table 2 provides details of these datasets. EMNIST-Balanced is a handwritten character dataset derived from the NIST database. CompCars contains car images collected from both network and surveillance scenarios. ImageNet 1K is a large-scale digital image dataset widely used in computer vision tasks. COCO is a large-scale dataset designed for image detection, semantic segmentation, and keypoint detection.

Experimental metrics

Speed-up ratio

The acceleration ratio (Maaz et al., 2022) is the time consumed by a task running in a single-node system and a multi-node system. It is often used to measure the acceleration performance and effectiveness of parallel systems, as defined below: (22) Sm=T1Tm

where m represents the number of nodes of the parallel system, T1 is the computing time of the task in a single-node system, and T2 is the parallel computing time of the task in a multi-node system with m nodes.

Top-1 accuracy

Top-1 accuracy 20 is the probability that the category with the highest probability in the model prediction results matches the actual category, and a higher value indicates better model classification performance, which is defined as follows: (23) Ftop−1=Ti/N

where Ti is the number of samples in the sample set that are correctly labeled in the best category label of the model output and N is the total number of samples.

Table 2 Experimental dataset details.

	Emnist-Balanced	CompCars	ImageNet1K	COCO	
Number	131,600	208,826	1,281,167	3,000,000	
Size	28 × 28	224 × 224	224 × 224	224 × 224	
Categories	47	1,716	1,000	91	

Experimental results and analysis

Visualization of feature map

To evaluate the impact of the PFE-MHO strategy on feature map accuracy, the feature representations of the algorithm were visualized to assess the necessity of applying the PEFS-MHO strategy to the VGG-16 model. The corresponding visualization results are presented in the following Figs. 2A and 2B.

Compared to the algorithm without the MHO-PEFS strategy, the version incorporating the PFE-MHO strategy produces feature maps with enhanced feature representation at each convolutional layer. This improvement is clearly illustrated in Fig. 2A, which demonstrates that the PFE-MHO strategy results in more structured and refined feature representations in each feature map. In contrast, Fig. 2B shows that without the PFE-MHO strategy, the boundary delineation of extracted features appears less distinct, particularly in areas of light and dark transitions.

The observed improvements can be attributed to the following factors: The PFE-MHO strategy enhances feature representation by employing non-local mean filtering based on cosine similarity and zero-crossing detection from the Laplacian operation. These techniques enable the extraction of edge features while effectively reducing noise interference. By improving the quality and accuracy of the data fed into the convolutional neural network, the strategy preserves essential information and provides a robust foundation for subsequent model training. This, in turn, contributes to improved model accuracy and stability.

Furthermore, the feature selection process within the PFE-MHO strategy optimizes model performance by eliminating redundant features. This is accomplished through the calculation of the feature correlation index and the application of a correlation coefficient threshold. By reducing data dimensionality, the strategy lowers computational resource consumption and minimizes the risk of overfitting, thereby enhancing the generalization capability of the model.

Figure 2 (A) Feature maps using the PFE-MHO strategy. (B) Feature maps without using the PFE-MHO strategy.

The benefits of this approach extend to predictive accuracy and classification performance when applied to new data. By ensuring more efficient and accurate model learning in complex data environments, the PFE-MHO strategy proves to be a highly effective method for enhancing feature extraction in the PDCNN-MI algorithm.

Algorithm feasibility

Experiments were conducted on four datasets: Emnist-Balanced, CompCars, ImageNet 1K, and COCO to assess the feasibility of the PDCNN-MI algorithm. Figure 3 illustrates the speedup performance of the algorithm across these datasets.

Figure 3 The speedup ratio of the PDCNN-MI algorithm.

As illustrated in Fig. 3, the speedup ratio for all four datasets—Emnist-Balanced, CompCars, ImageNet1K, and COCO—demonstrates a consistent increase as the number of computational nodes grows, peaking when the node count reaches eight. Specifically, with two nodes, the speedup ratios are 1.61, 1.54, 1.75, and 1.63, respectively. When the number of nodes is doubled to four, the speedup ratios rise significantly to 2.92, 3.08, 3.3, and 3.17. Further increasing the node count to six results in even higher speedup ratios of 4.26, 4.03, 4.67, and 4.46. Finally, with eight nodes, the speedup ratios reach their maximum values of 5.1, 5.4, 5.85, and 6.2, indicating that the algorithm achieves its highest efficiency at this node count. This trend underscores the algorithm’s ability to effectively leverage additional computational resources to accelerate performance across diverse datasets.

Additionally, the PDCNN-MI algorithm demonstrates a more significant speed improvement when processing large-scale datasets such as ImageNet 1K and COCO. This observation demonstrates the efficiency of the algorithm in handling large volumes of data, confirming its scalability and effectiveness in high-performance computing environments.

Algorithm runtime analysis

To evaluate the runtime efficiency of the proposed algorithms in a large-scale data environment, experiments were conducted to compare the performance of the N-SAMCN, ECSO-FRCNN, and MR-LSCNN algorithms with the PDCNN-MI algorithm. The comparison criterion was the total runtime required for each algorithm to achieve stable training accuracy on the specified dataset. The experimental results are illustrated in Fig. 4.

Figure 4 The running time of each algorithm.

As shown in Fig. 4, the difference in training time between the PDCNN-MI algorithm and the other algorithms is minimal when processing small datasets such as EMNIST-Balanced and CompCars. Specifically, the PDCNN-MI algorithm reduces training time by 28.76 min, 15.21 min, and 37.84 min for EMNIST-Balanced and by 35.87 min, 8.45 min, and 45.93 min for CompCars, compared to the other algorithms. However, the training time difference becomes more substantial when dealing with large-scale datasets such as ImageNet 1K and COCO. In ImageNet 1K, the PDCNN-MI algorithm reduces 91.64 min, 56.39 min, and 186.12 min, while in COCO, the reductions are 106.73 min, 65.28 min, and 197.51 min. These results demonstrate that the PDCNN-MI algorithm significantly improves computational efficiency, particularly for large-scale data processing.

The computational inefficiency of N-SAMCN, ECSO-FRCNN, and MR-LSCNN on large-scale datasets primarily arises from their algorithmic deficiencies in three key aspects: inadequate handling of feature redundancy in high-dimensional data leading to computational waste, insufficient optimization of convolutional operations for high-resolution inputs, and defective convergence mechanisms due to poorly designed loss functions, suboptimal optimization algorithms, and ineffective handling of data imbalance. These fundamental shortcomings collectively result in significantly prolonged runtime during both training and inference phases.

The reasons why the PDCNN-MI algorithm performs well are: The PMT-IM strategy calculates the covariance matrix and mean of all convolutional kernels in the convolutional layer at each node, constructing the objective function for the Mahalanobis distance center value (MDCV). By computing the second-order Taylor expansion of the objective function at its stationary points, the strategy iteratively identifies the optimal MDCV value. This process removes redundant convolutional kernels that are highly correlated with others in the current convolutional layer, thereby reducing redundant parameters in the network model, decreasing computational load and storage requirements, and ultimately accelerating convolutional layer operations. Additionally, by integrating the Im2col algorithm with the MapReduce computational framework, the strategy implements parallel Im2col convolution, transforming convolution operations into matrix multiplications and executing them in parallel across multiple computational nodes. This approach significantly enhances the computational speed of the convolutional layer. Compared to traditional convolution methods, the PMT-IM strategy can substantially reduce model training time and improve the overall performance of the algorithm.

Speedup ratio analysis

To assess the parallel computing performance of the PDCNN-MI algorithm in a large data environment, 10 tests were conducted on the above four data sets for the N-SAMCN, MR-LSCNN, and ECSO-FRCNN algorithms. The mean values from these tests were used to compute the speedup ratios of each algorithm across different numbers of computational nodes. The experimental results are presented in Fig. 5.

Figure 5 The speedup ratio of each algorithm.

As shown in Fig. 5, the speedup ratios of all algorithms across the four datasets increase as the number of computational nodes grows. The speedup ratio gap between the algorithms is relatively small when processing EMNIST-Balanced, a smaller dataset, as illustrated in Fig. 5A. When the number of nodes is two, the speedup ratio of the PDCNN-MI algorithm is 0.06, 0.01 and 0.21 higher than that of the N-SAMCN, ECSO-FRCNN and MR-LSCNN algorithms, as the number of nodes increases to eight, the speedup ratio of the PDCNN-MI algorithm exceeds those of the other algorithms by 0.42, 0.57, and 0.69, respectively. However, when processing COCO, a dataset with a significantly larger data size, the speedup ratio gap between the algorithms becomes more pronounced, as depicted in Fig. 5D. When the number of nodes is two, the PDCNN-MI algorithm achieves speedup ratios of 0.01, 0.03, and 0.17. higher than the N-SAMCN, ECSO-FRCNN and MR-LSCNN algorithms. As the number of nodes increases to eight, the PDCNN-MI algorithm outperforms the other algorithms with speedup ratios of 0.48, 0.39, and 1.32 higher, respectively.

The reasons for the slowdown in the speedup ratio of N-SAMCN, ECSO-FRCNN, and MR-LSCNN algorithms as the number of parallel nodes increases are as follows: although these algorithms employ parallel frameworks combined with DCNNs for parallel computation, they fail to account for the multi-dimensional characteristics of big data environments. Consequently, the efficiency improvement of convolutional operations becomes less significant as the data scale grows and the number of parallel nodes increases. As a result, when the number of parallel nodes rises, the trend of parallel efficiency growth diminishes.

These results demonstrate that the PDCNN-MI algorithm exhibits superior parallel computing efficiency, mainly when processing large-scale datasets. This improvement can be attributed to two key factors. First, the PMT-IM strategy optimizes convolutional operations by calculating the covariance matrix and mean of all convolutional kernels in the convolutional layer at each computational node. It then constructs an objective function for the MDCV. By computing the second-order Taylor expansion of this objective function at its stationary points, the strategy iteratively determines the optimal MDCV value. This process effectively removes redundant convolutional kernels that correlate highly with others in the current convolutional layer. As a result, the strategy reduces redundant parameters in the network model, lowers computational load and storage requirements, and accelerates convolutional layer operations. Second, integrating the Im2col algorithm with the MapReduce computational framework enables parallel Im2col convolution. This transformation converts convolution operations into matrix multiplications, executed in parallel across multiple computational nodes. By leveraging this approach, the strategy significantly enhances the computational efficiency of the convolutional layer. Compared to traditional convolution methods, the PMT-IM strategy substantially reduces model training time while improving the overall performance of the algorithm. These optimizations collectively demonstrate the effectiveness of PDCNN-MI in handling large-scale datasets with improved speed and efficiency.

Classification effect analysis

To evaluate the classification performance of the PDCNN-MI algorithm, a comparative analysis was conducted against the N-SAMCN, MR-LSCNN, and ECSO-FRCNN algorithms across four datasets. The evaluation was based on Top-1 accuracy and floating-point operations per second (FLOPs) as performance metrics. The experimental results are presented in Table 3.

As shown in Table 3, the Top-1 accuracy of each algorithm is notably higher when applied to small-scale datasets such as EMNIST-Balanced. Specifically, the Top-1 accuracy of PDCNN-MI reaches an impressive 95.21%, outperforming N-SAMCN, MR-LSCNN, and ECSO-FRCNN by 2.90%, 4.11%, and 2.20%, respectively. Concurrently, PDCNN-MI demonstrates superior computational efficiency, with its FLOPs reduced by 6.7%, 29.96%, and 19.7% compared to N-SAMCN, MR-LSCNN, and ECSO-FRCNN, respectively. This highlights PDCNN-MI’s ability to achieve higher accuracy while requiring fewer computational resources on smaller datasets. The performance gap becomes even more pronounced when handling large-scale datasets like COCO. On COCO, PDCNN-MI achieves a Top-1 accuracy of 88.91%, surpassing N-SAMCN, MR-LSCNN, and ECSO-FRCNN by 3.64%, 4.89%, and 7.39%, respectively. Moreover, PDCNN-MI maintains its computational efficiency advantage, with FLOPs reduced by 14.29%, 42.42%, and 26.26% compared to N-SAMCN, MR-LSCNN, and ECSO-FRCNN, respectively. This demonstrates that PDCNN-MI not only delivers superior accuracy but also scales effectively to larger datasets with significantly lower computational costs.

The lower accuracy and higher computational costs of N-SAMCN, MR-LSCNN, and ECSO-FRCNN stem from their inability to effectively address poor loss function convergence in large-scale datasets. Challenges like vanishing gradients, overfitting, and inefficient parameter updates hinder their optimization. These algorithms likely lack advanced techniques such as adaptive learning rates or robust regularization, leading to suboptimal performance.

Table 3 Top 1 of each algorithm on four datasets.

Dataset	Algorithm	Top1 Acc	FLOPs	
Emnist-Balanced	PDCNN-MI	95.21%	7.31 ×106	
MR-LSCNN	92.45%	9.50 ×106	
ECSO-FRCNN	91.30%	8.75 ×106	
N-SAMCN	93.12%	7.80 ×106	
CompCars	PDCNN-MI	93.81%	1.63 ×107	
MR-LSCNN	90.56%	2.22 ×107	
ECSO-FRCNN	89.45%	1.95 ×107	
N-SAMCN	91.20%	1.70 ×107	
ImageNet 1K	PDCNN-MI	89.38%	6.21 ×109	
MR-LSCNN	85.12%	8.37 ×109	
ECSO-FRCNN	83.45%	7.10 ×109	
N-SAMCN	86.78%	6.50 ×109	
COCO	PDCNN-MI	88.91%	1.98 ×1010	
MR-LSCNN	84.56%	2.82 ×1010	
ECSO-FRCNN	82.34%	2.50 ×1010	
N-SAMCN	85.67%	2.31 ×1010	

The reasons why the PDCNN algorithm performs excellently are: First, the IMBGD strategy refines gradient calculation by leveraging batch data loss analysis and distributed computing. Initially, it computes the mean loss of the batch using the defined loss function formula. The loss mean weight is then determined by evaluating the difference between each data point’s loss function value and the mean loss. Following this, the partial derivatives of the loss function for each data point with respect to the current model parameters are computed. These derivative values are then mapped into key-value pairs and stored in HDFS for efficient processing. Using these key-value pairs as indices, the method constructs the batch data average gradient, represented as LSG. This strategy enhances the gradient estimation process, ensuring that the calculated batch gradient better represents the true gradient direction of the data. By mitigating the influence of outliers, which could introduce deviations in the gradient direction, the IMBGD approach improves the precision of parameter updates, leading to more stable and accurate model optimization.

In addition, in big data environments, distributed clusters often encounter anomalies such as program interruptions and memory overflows, leading to unstable loss function behavior during backpropagation and poor convergence. The IMBGD strategy addresses this challenge by designing the loss mean weight to exclude the impact of abnormal node training data on the batch gradient. This mechanism stabilizes loss function convergence, preventing oscillations caused by noisy data and improving the overall stability of the algorithm. The strategy ensures that parameter updates proceed efficiently toward the optimal solution by constructing the batch data average gradient LSG and accurately computing the gradient direction.

Applications

The PDCNN-MI algorithm represents a significant breakthrough in landslide prediction and monitoring, overcoming the limitations of conventional DCNN in processing complex and noisy remote sensing data. By integrating the parallel feature extraction strategy based on the Marr-Hildreth Operator (PFE-MHO), the algorithm effectively isolates key landslide indicators such as slope gradient, surface cracks, and terrain deformation while filtering out extraneous elements like vegetation and cloud cover. This refinement improves landslide detection accuracy and enhances computational efficiency, making it viable for real-time applications.

Furthermore, the parallel model training strategy based on the Im2col method (PMT-IM) optimizes the convolution process, enabling the rapid extraction of spatial features crucial for analyzing landslide features. The Improved Mini-Batch Gradient Descent Strategy (IMBGD) further strengthens the robustness of the model by mitigating the impact of anomalous data, ensuring stable and reliable training outcomes.

Beyond landslide detection, PDCNN-MI’s adaptability extends to other geological hazards including debris flows and collapses, demonstrating its versatility and potential to transform disaster prevention and mitigation strategies at a global scale. In an era of increasing susceptibility to natural disasters, such advancements offer invaluable tools for fostering safer and more resilient communities worldwide.

Conclusions

This article presents a novel model for remote sensing image recognition in big data environments, addressing the challenges of excessive redundant features, slow convolution operations, and poor loss function convergence. The proposed approach integrates a feature extraction strategy based on the Marr-Hildreth operator to minimize data redundancy, a model training strategy that combines the Im2col method and Mahalanobis distance center value to eliminate redundant convolutional kernels, and an improved gradient descent strategy designed to exclude abnormal node influences and ensure robust convergence.

Experimental evaluations on the EMNIST-Balanced, CompCars, ImageNet 1K, and COCO datasets demonstrate superior accuracy and computational efficiency compared to existing methods. The parallel training framework effectively manages the complexities of big data while preserving global feature integrity, making it particularly suitable for applications such as landslide prediction.

To further enhance the performance and applicability of the proposed model, several research directions will be explored. Addressing class imbalance d in real-world remote sensing datasets will improve classification accuracy and robustness. Optimizing dynamic resource allocation will enhance computational efficiency by adapting resource distribution based on data complexity and task requirements. Extending the model to support real-time processing will enable faster decision-making in critical scenarios such as disaster response. Additionally, exploring cross-domain adaptability will allow the model’s transferability to other fields, such as medical imaging and autonomous driving, to validate its generalization capability and robustness.

These advancements will ensure the proposed model remains a versatile and effective solution for big data-driven remote sensing applications while expanding its potential for broader real-world use cases.

Supplemental Information

Supplemental Information 1 Source Code

Additional Information and Declarations

Competing Interests

Author Contributions

Data Availability

The authors declare there are no competing interests.

Yimin Mao conceived and designed the experiments, performed the experiments, analyzed the data, performed the computation work, prepared figures and/or tables, authored or reviewed drafts of the article, and approved the final draft.

Yaser Ahangari Nanehkaran performed the experiments, prepared figures and/or tables, authored or reviewed drafts of the article, and approved the final draft.

Neelakandan Chandrasekaran analyzed the data, prepared figures and/or tables, authored or reviewed drafts of the article, and approved the final draft.

Ying Huo analyzed the data, prepared figures and/or tables, authored or reviewed drafts of the article, and approved the final draft.

Zhan Qing Wen analyzed the data, prepared figures and/or tables, authored or reviewed drafts of the article, and approved the final draft.

ke Gong analyzed the data, prepared figures and/or tables, authored or reviewed drafts of the article, and approved the final draft.

Miao Decheng performed the experiments, prepared figures and/or tables, authored or reviewed drafts of the article, and approved the final draft.

The following information was supplied regarding data availability:

The PDCNN-MI code is available at GitHub: https://github.com/hhpcdnn/PDCNN-MI.git.

The EMNIST Dataset is available at the National Institute of Standards and Technology: https://www.nist.gov/itl/products-and-services/emnist-dataset.

The Comprehensive Cars (CompCars) dataset is available at:

https://mmlab.ie.cuhk.edu.hk/datasets/comp_cars.

ImageNet Data is available at https://www.image-net.org/download.php.

The Coco dataset is available at: https://cocodataset.org/#download.

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
