# Peer review of "An efficient parallel DCNN algorithm in big data environment"

_PeerJ Computer Science, doi:10.7717/peerj-cs.2871_

## Round 0.1 · original submission · Major Revisions

Dear Authors,

Your paper has been reviewed. Based on the reviewers' reports, major revisions are needed before it is considered for publication in PEERJ Computer Science. The issues you have to fix in your revised version of your paper are mainly the following:

1) you must demonstrate how PDCNN-MI mitigates the identified DCNN challenges and its suitability for big data image processing environments.

2) You must improve the clarity of the section Introduction by adding the latest relevant literature review to improve understanding and quality.

3) You must compare the performance of your method with a state-of-the-art method that is relevant to the big data environment, clarifying how you extract the features

·

Basic reporting

Author has provided sufficient background information and literature review regarding an efficient parallel DCNN algorithm in big data environment. The PDCNN-MI 139 algorithm is proposed to solve the problem of excessive redundant features, slow convolution operation, and poor loss function convergence in the process of remote sensing image classification.

Experimental design

In this paper, the author has designed the PMT-IM strategy to prune same-layer convolution kernels and has also used MapReduce and Im2col methods for parallel training to increase the speed of convolutional layer operations.

Moreover, in the stage of parameter parallel updating, the author has proposed the IMBGD strategy to exclude the influence of abnormal nodes during training data on batch gradient descent, effectively addressing the poor convergence of the loss function.

Validity of the findings

In this paper, PDCNN-MI algorithm, both strategies of PFE-MHO and IMBGD are proposed to address the problems of parallel DCNN algorithms in big data environments to filter the input data by the improved non-local mean filter, which solves the problem of high data redundancy.

Additional comments

The author has illustrated the issues, problems, and trends related to strategies, algorithms, techniques, and simulations aimed at solving critical issues in the big data environment.

Cite this review as

Reviewer 2 ·

Basic reporting

By structuring the content this way, you can clearly demonstrate how PDCNN-MI mitigates the identified DCNN challenges and its suitability for big data image processing environments. This will effectively communicate the technical advancements and practical relevance of the algorithm.Introduction section is not well written for the reader understanding and also add the latest relevent literture review to better understanding and quality.

Experimental design

In result and discussion , author didnot compared with state of art method which is relevent of the bigdata environment. how to extract the feature? DCNNs can suffer from feature redundancy, which leads to inefficiencies in storage and processing. how to the improvement in classification accuracy and processing speed when compared to traditional DCNNs. Emphasize that PDCNN-MI’s architecture is optimized for high efficiency, making it particularly suitable for fast, large-scale image processing tasks.

Validity of the findings

how to validate your system?. To further refine technical details and emphasize practical applications.

Additional comments

In this paper no novality exist and result s are not varified according the research.

Cite this review as

Reviewer 3 ·

Basic reporting

1- Abstract: At the end state of the significance of the obtained results (highest value)

2- The introduction should discuss the topic in detail and include research questions and targets.

Experimental design

1 Add a new section for training and validating the models.

Validity of the findings

1. Major findings and important outcomes should be addressed in the conclusion, not in the introduction.

Additional comments

1- The introduction can be improved through clarity and organization. Consider breaking the introduction, more focused paragraphs, each addressing a specific aspect. This will help readers better follow the logical progression of your introduction. Also, separate the introduction from related work make it two sections. Introduction: It would be better to explicitly state the research questions of this study and in the discussion section, explain how you achieved these research questions using the applied methodology and the obtained results. Also, Highlight the novel contribution of this work, compared to the existing latest related studies.
2- Highlight the novel contribution of your work, compared to the existing latest related studies.

3- Justify the selection of your model for the comparative study.
4- Emphasize the novel scientific contribution of your work.
5- What is the generalizability of your model. Does it support other datasets as well?

Cite this review as

·

Basic reporting

1. The manuscript should be proofread further, to improve the presentation clarity. (The intended meaning often lacks clarity)
2. The proposed solution is briefly described. A lot of information should be given here about the consideration of the developed architecture.
3. How do you deal with dataset imbalance problems? Did you consider any data augmentation techniques?
4. Discuss the limits of the proposed approach and the validity of experimental results.
5. There are generalized statements and hence author needs to identify precisely the problem they wish to solve.
6. Results need to be briefly discussed.
7. Overall, The author must highlight the scientific contribution, describe better his/her approach, and give enough information to compare the results.

Experimental design

As above

Validity of the findings

As above

Additional comments

As above

Cite this review as

---

## Round 0.2 · Major Revisions

Dear Authors,

Your paper has been revised. It needs major revisions before being considered for publication in PEERJ Computer Science. More precisely:

1) The manuscript should be proofread further to improve its presentation clarity. To this end, you must highlight the scientific contribution, describe the proposed approach better, and give enough information to compare the results.

2) You must describe how you deal with dataset imbalance problems. Did you consider any data augmentation techniques?

·

Basic reporting

see Additional comments

Experimental design

see Additional comments

Validity of the findings

see Additional comments

Additional comments

1. The manuscript should be proofread further, to improve the presentation clarity. (The intended meaning often lacks clarity)
2. The proposed solution is briefly described. A lot of information should be given here about the consideration of the developed architecture.
3. How do you deal with dataset imbalance problems? Did you consider any data augmentation techniques?
4. Discuss the limits of the proposed approach and the validity of experimental results.
5. There are generalized statements and hence author needs to identify precisely the problem they wish to solve.
6. Results need to be briefly discussed.
7. Overall, The author must highlight the scientific contribution, describe better his/her approach, and give enough information to compare the results.
8. Further Papers can be read and used

Cite this review as

---

## Round 0.3 · accepted · Accept

Dear Authors,
Your paper has been revised. It has been accepted for publication in PEERJ Computer Science. Thank you for your fine contribution.

·

Basic reporting

Good

Experimental design

Good

Validity of the findings

Good

Cite this review as